# Generalized Contrastive Learning for Universal Multimodal Retrieval

**Jungsoo Lee   Janghoon Cho   Hyojin Park   Munawar Hayat**
**Kyuwoong Hwang   Fatih Porikli   Sungha Choi**[†]

Qualcomm AI Research[*]
{jungsool, janghoon, hyojinp, hayat, kyuwoong, fporikli, sunghac}@qti.qualcomm.com

## Abstract

Despite their consistent performance improvements, cross-modal retrieval models (*e.g.*, CLIP) show degraded performances with retrieving keys composed of fused image-text modality (*e.g.,* Wikipedia pages with both images and text). To address this critical challenge, multimodal retrieval has been recently explored to develop a unified single retrieval model capable of retrieving keys across diverse modality combinations. A common approach involves constructing new composed sets of image-text triplets (*e.g.*, retrieving a pair of image and text given a query image). However, such an approach requires careful curation to ensure the dataset quality and fails to generalize to unseen modality combinations. To overcome these limitations, this paper proposes Generalized Contrastive Learning (GCL), a novel loss formulation that improves multimodal retrieval performance without the burdensome need for new dataset curation. Specifically, GCL operates by enforcing contrastive learning across all modalities within a mini-batch, utilizing existing image-caption paired datasets to learn a unified representation space. We demonstrate the effectiveness of GCL by showing consistent performance improvements on off-the-shelf multimodal retrieval models (*e.g.*VISTA, CLIP, and TinyCLIP) using the M-BEIR, MMEB, and CoVR benchmarks.

## 1   Introduction

With the growing availability of multimodal data, the ability to retrieve relevant keys across different modalities has become increasingly important. While cross-modal retrieval, retrieving images with given text or vice versa, has garnered significant attention and progress [1, 2, 3, 4, 5, 6, 7, 8], performing retrieval with fused image-text modality still remains a challenge [9, 10]. Consider, for instance, the task of finding a Wikipedia page composed of both images and text (*e.g.,* the Eiffel Tower paired with its history) in response to a query (*e.g.,* "What is the history of the Eiffel Tower?"). For such real-world multimodal retrieval scenarios, existing approaches deliver limited performance.

This performance drop stems primarily from the pervasive modality gap [11, 12, 13, 14, 15, 16, 17], a critical barrier in retrieval systems. The modality gap arises when semantically similar samples across different modalities (*e.g.,* an image and its caption) exhibit low similarity in the embedding space, while semantically dissimilar samples within the same modality appear misleadingly close [18]. For example, although an image of teddy bears is paired with the annotated caption 'a photo of teddy bears', it may be embedded closer to images of other animals in the representation space, rather than to its corresponding caption. This misalignment becomes especially acute when retrieval keys - spanning text-only, image-only, or fused text-image formats - are stored in a unified database [9, 10]. For example, a search query might ideally match a Wikipedia page with both images and text. However, without a shared representation space to bridge the three different modalities, models struggle to pinpoint the most relevant candidate, which undermines their effectiveness.

---

[*]Qualcomm AI Research is an initiative of Qualcomm Technologies, Inc. [†] indicates corresponding author.

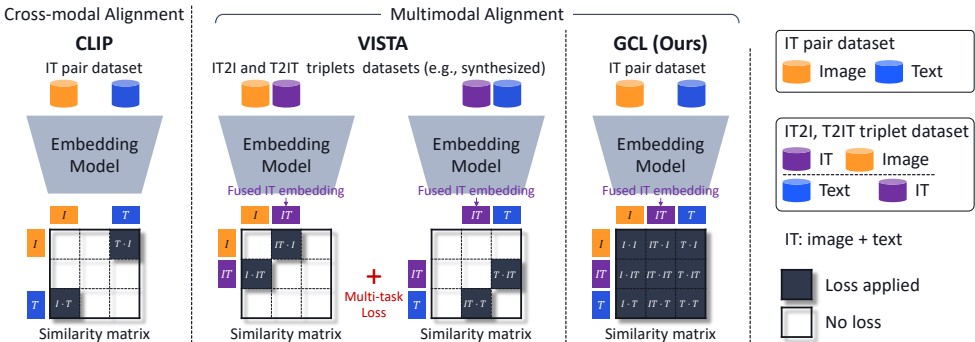

Figure 1: Overview of GCL. Given an embedding model pretrained for cross-modal alignment, previous studies (*e.g.*, VISTA [19]) constructed new triplet datasets to simulate specific multimodal retrieval scenarios. However, this approach limits generalization to unseen retrieval scenarios (white squares). In contrast, GCL improves retrieval performance across diverse scenarios (black squares). Specifically, by utilizing off-the-shelf image-caption datasets, GCL enables the learning of retrieval tasks involving nine different modality combinations.

To tackle this issue, recent research has extensively studied multimodal retrieval, which extends cross-modal retrieval by allowing searches across various modality combinations, including data samples that contain both images and text [19, 20, 9, 21, 22, 23, 24, 25, 10, 26]. As demonstrated in Fig. 1, previous studies have attempted to improve multimodal retrieval performance by generating specialized datasets tailored to specific retrieval scenarios [19, 20]. For example, VISTA [19] finetuned a pretrained cross-modal retrieval model with newly generated datasets composed of triplets: 1) IT2I dataset which includes queries of image-text pairs with candidate images and 2) T2IT dataset which includes query text with candidates of image-text pairs. While this approach can be effective for the targeted retrieval scenarios, it requires meticulous curation to ensure the quality of the generated samples (*e.g.,* verifying that the generated images accurately represent the intended content). Moreover, models trained on these composed datasets may fail to generalize to unseen modality combinations beyond those encountered in the training sets (*e.g.*, retrieving fused text-image samples when provided with corresponding text-image queries). Fig. 1 shows that the model only learns the retrieval scenarios included in the datasets (black squares), leaving the other combinations (white squares) unlearned during the finetuning phase. While such studies resort to generating new datasets for learning specific retrieval scenarios, little has been explored to fully utilize off-the-shelf image-caption paired datasets for improving multimodal retrieval performances.

To this end, we propose Generalized Contrastive Learning (GCL), a *simple yet effective* loss function that enhances multimodal retrieval performance by leveraging existing image-caption paired datasets, sidestepping the need for costly dataset construction. Specifically, GCL integrates three types of embeddings - text embeddings, image embeddings, and fused text-image embeddings - and applies contrastive loss across all modalities within a mini-batch to learn a unified representation space. As shown in Fig. 1, GCL encourages positive pairs from different modalities to be pulled closer together while pushing apart all negative pairs, regardless of modalities. This training process enables the retrieval model to learn retrieval between all combinations of modalities, which was limited to certain pairs in the previous methods that generated specific triplets. Despite its simplicity, GCL consistently improves multimodal retrieval performances across diverse tasks and datasets and outperforms a model trained with newly composed triplet datasets. The key advantage of GCL is that it does not require expensive dataset curation and can generalize well to various multimodal retrieval scenarios.

The major contributions of this paper are:

- We propose Generalized Contrastive Learning (GCL), a novel contrastive learning approach that improves multimodal retrieval by integrating different modalities (text, image, and fused text-image) within a mini-batch for building a unified representation space.

- Unlike previous methods that rely on expensive and manually curated composed datasets, GCL effectively leverages existing image-caption paired datasets, making it a cost-efficient and scalable solution for multimodal retrieval.

- We show that GCL significantly enhances multimodal retrieval performance on diverse benchmarks (*e.g.*, M-BEIR [9], MMEB [27], and CoVR [28]) and retrieval models (*e.g.*, VISTA [19], CLIP [1], and TinyCLIP [4]), showing its broad applicability and effectiveness.

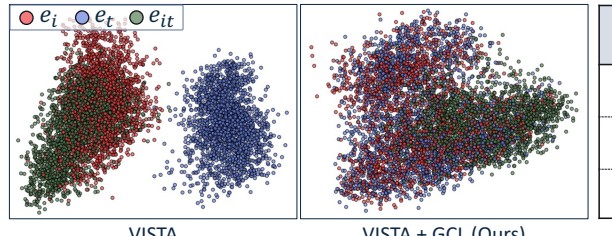

| | Cosine similarity between averaged vectors of each modality | | |
| Modality Pair | $\bar{e}_i \leftrightarrow \bar{e}_t$ | $\bar{e}_i \leftrightarrow \bar{e}_{it}$ | $\bar{e}_t \leftrightarrow \bar{e}_{it}$ |
|---|---|---|---|
| VISTA | 0.8799 | 0.7393 | 0.6772 |
| **VISTA + GCL (Ours)** | **0.8882** | **0.9688** | **0.8740** |

Figure 2: PCA visualization of representation spaces using $e_i$, $e_t$, and $e_{it}$. We use MSCOCO for $e_i$ (red) and $e_t$ (blue) and WebQA for $e_{it}$ (green). We sampled 2K samples from each modality, using 6K samples in total. $\bar{e}$ indicates the average embedding vector of each modality.

## 2 Related Work

### 2.1 Cross-modal Retrieval and Contrastive Learning

Cross-modal retrieval has been widely studied to enable searches across different modalities, such as retrieving images based on text queries or vice versa [1, 11, 2, 4, 3, 6, 12, 13, 8]. By leveraging contrastive learning with large-scale image-text pairs, both image and text embeddings are mapped into a shared representation space [1]. However, retrieval models trained in this manner still suffer from the modality gap, a discrepancy between image and text embeddings even with the same semantics [12, 13, 15, 16, 17, 18].

To address this issue, recent studies have proposed diverse techniques to reduce the modality gap [12, 11]. For instance, AlignCLIP introduces an intra-modality separation loss, which pushes apart samples within the same modality to improve cross-modal alignment [11]. While this approach helps mitigate the modality gap, it is mainly designed for cross-modal retrieval and does not explicitly handle data samples that contain both images and text (*e.g.*, social media pages with both images and text descriptions). As a result, its effectiveness in multimodal retrieval scenarios remains limited.

### 2.2 Multimodal Retrieval

Multimodal retrieval builds upon cross-modal retrieval by supporting searches across different combinations of modalities, including samples that incorporate both images and text [19, 20, 9, 29, 30, 31, 21, 22, 23, 32, 33]. UniIR demonstrates that adding image and text embeddings is effective for fusing representations for CLIP-based models, termed score-fusion (SF) [9]. The fused embeddings enable the retrieval of data samples that contain both images and text.

As mentioned previously, one common approach for multimodal retrieval is generating composed datasets containing paired image and text samples tailored for specific retrieval scenarios. For example, by using image generation models, VISTA generated the IT2T dataset, which consists of image and text queries with text-based keys, and T2IT dataset, which includes text queries with both image and text keys [19]. Similarly, MegaPairs constructed triplets consisting of two images and a descriptive text capturing the relationship between them, which are generated with multimodal large language models [20]. While dataset generation through large generative models can be effective under certain scenarios, it requires careful curation to ensure data quality, making the process labor-intensive and computationally expensive. Additionally, the retrieval model trained with such generated datasets may fail to generalize to scenarios unseen during the training phase. Therefore, a more efficient and scalable approach is needed to enhance multimodal retrieval performance without relying on expensive dataset generation.

## 3 Method

### 3.1 Problem Setup

In this paper, we define $(x_i, x_t)$ as a pair of image and text used to train a retrieval model $\theta$, which consists of an image encoder $\theta_i$ and a text encoder $\theta_t$. The extracted embeddings of images and text are denoted as $e_i = \theta_i(x_i)$ and $e_t = \theta_t(x_t)$, respectively. During inference, the model retrieves candidate samples $c$ when given a query $q$.

In cross-modal retrieval, the model retrieves text candidates $c_t$ for an image query $q_i$ ($q_i \rightarrow c_t$) and vice versa ($q_t \rightarrow c_i$). The multimodal retrieval task generalizes this setting by incorporating samples that contain both images and text, denoted as $x_{it}$. This results in more complex retrieval scenarios,

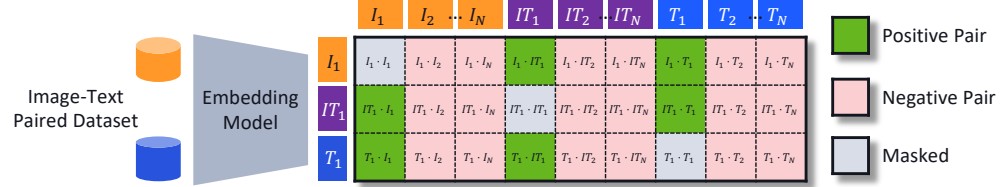

Figure 3: Training process of GCL. Given a dataset composed of image-caption pairs, we extract $e_i$, $e_t$, and $e_{it}$. For $e_{it}$, we follow the extraction method used by the retrieval model (*e.g.*, VISTA and CLIP-SF). Then, we integrate samples of the three different modalities into a single mini-batch for contrastive learning. We mask out the supervision on the positive samples with identical modalities.

where candidates of arbitrary modalities are retrieved based on queries of arbitrary modalities, such as $q_t \rightarrow c_{it}$, $q_{it} \rightarrow c_i$, and $q_{it} \rightarrow c_{it}$. Following UniIR [9] we consider two different retrieval settings: (1) global setting, retrieving candidates from a shared database regardless of modalities and tasks, and (2) local setting, retrieving candidates from a task-specific database with the same modality. We conduct experiments on both settings in this paper.

Figure 2 clearly illustrates the challenge addressed in this work. The left side of Figure 2 presents a PCA [34] visualization of the embedding spaces learned by VISTA and VISTA fine-tuned with our proposed loss function, Generalized Contrastive Learning (GCL). For this visualization, we use $c_i$ and $c_t$ from MSCOCO and $c_{it}$ from WebQA. As shown, VISTA fails to construct a unified embedding space for the three different modalities. The main reason is that retrieval models trained on image-caption paired datasets [2] fail to learn a shared embedding space that incorporates $e_{it}$.

The table on the right further supports this observation. By using the same samples from the visualization, we compute the average embeddings of images, text, and image-text pairs, denoted as $\overline{e}_i$, $\overline{e}_t$, and $\overline{e}_{it}$, respectively. We then calculate the cosine similarity between the average embedding vectors of each modality. As demonstrated in the table, VISTA exhibits low cosine similarity across modalities, indicating a significant modality gap. Our goal is to mitigate this gap, as evidenced by the more intermixed scatter plots in the PCA visualization and the increased cosine similarities between modalities after finetuning with GCL.

## 3.2 Generalized Contrastive Learning

Using $N$ samples per mini-batch, the standard contrastive learning loss is formulated as:

$$\mathcal{L}_{\text{CL}} = -\frac{1}{2N} \sum_{j=1}^{N} \sum_{(a,b) \in S} \log \frac{\exp[(e_a^j \cdot e_b^j)/\tau]}{\sum_{k=1}^{N} \exp[(e_a^j \cdot e_b^k)/\tau]}, \tag{1}$$

where $j$ and $k$ denote the sample indices, $\tau$ is the temperature scaling parameter, and $S$ denotes the set of modality pairs $\{(i,t), (t,i)\}$. While this loss function has been effective in constructing a unified cross-modal representation space, multimodal retrieval performance remains limited within this embedding space. The main limitation arises because models are not explicitly trained with embeddings that represent data samples containing both images and text (*e.g.*, news articles with both pictures and textual explanations). As mentioned earlier, previous studies created new datasets consisting of triplets to simulate specific retrieval scenarios for addressing such a limitation [19, 20]. In contrast, our proposed method does not rely on these new triplet-based datasets.

Fig. 3 describes our method well. Generally, multimodal retrieval models are obtained by finetuning cross-modal retrieval models, which are pretrained with image-caption paired datasets. Instead of finetuning the cross-modal retrieval models with newly constructed triplet-based datasets, we finetune them using off-the-shelf image-caption paired datasets with our proposed Generalized Contrastive Learning (GCL) loss function. In GCL loss, negative samples are constructed from all possible combinations of embeddings (pink squares), including 1) image embeddings $e_i$, 2) text embeddings $e_t$, and 3) fused embeddings of images and text $e_{it}$. To obtain the fused embeddings of samples with both images and text, we follow the extraction method used in the retrieval model. For example, we either (1) use a specialized architecture pretrained for extracting fused embeddings for VISTA

---

[2]For the experiment, we use the checkpoint of VISTA prior to training with newly generated composed sets.

(*e.g.,* appending visual tokens alongside text tokens as input to a text encoder [19]), or (2) sum the individual image and text embeddings, $e_{it} = e_i + e_t$ for CLIP-based models following UniIR [9]. The positive pairs are then defined as samples with different modalities from the same pair (green squares). Positive pairs with the same modality correspond to the sample itself, so they are masked out during training (gray squares). Using the image embedding query as an example, the positive pair can be either the corresponding text embedding or the fused image-text embedding. GCL loss can be formulated as:

$$\mathcal{L}_{\text{GCL}} = -\frac{1}{6N} \sum_{j=1}^{N} \sum_{(a,b) \in P} \log \frac{\exp[(e_a^j \cdot e_b^j)/\tau]}{\sum_{m \in M} \sum_{k=1}^{N} \exp[(e_a^j \cdot e_m^k)/\tau]}, \tag{2}$$

where $M$ represents the set of modalities $\{i, t, it\}$ and $P$ denotes the set of positive modality pairs $\{(i,t), (i,it), (t,i), (t,it), (it,t), (it,i)\}$. The factor $6N$ normalizes the total loss across all six modality combinations in $P$, with each combination contributing losses from $N$ training samples.

Although AlignCLIP [11] recently introduced an intra-modality separation loss to push negative image samples away from a given image query in order to reduce the modality gap, it does not incorporate contrastive learning across all possible combinations, particularly those involving fused embeddings $e_{it}$. In contrast, our GCL loss provides a more generalized contrastive learning framework by seamlessly integrating different modalities into a unified representation space within a single mini-batch, leading to improved multimodal retrieval performances across diverse scenarios and tasks (results shown in Table 5).

## 4 Experiments

### 4.1 Experimental Settings

**Benchmarks** We evaluate the effectiveness of our proposed GCL using standard multimodal retrieval benchmarks: M-BEIR[9], MMEB[27], and CoVR [28] [3]. While images are used as input for M-BEIR and MMEB, CoVR handles video input. For M-BEIR, we conduct evaluations on 10 datasets under both local and global evaluation settings. For MMEB, we perform experiments on 12 sub-datasets included in the retrieval benchmark. Regarding CoVR, following the original evaluation setting of CoVR, we sample 15 frames for each target video and average the embeddings of each frame to obtain a single visual embedding for a given video. For the image-caption paired dataset used during finetuning, we use the LLaVA Visual Instruct Pretrain LCS-558K dataset [35], in which personal information has been blurred for data sanitization. Note that our experiments are conducted in a zero-shot setting, meaning the model is not fine-tuned on the training set of the evaluation benchmark.

**Models** We apply the GCL loss to recent multimodal retrieval models, VISTA [19], CLIP [1], and TinyCLIP [4] [4]. For VISTA, we use the checkpoint before the second stage, the one trained without the generated dataset to demonstrate that our method can enhance multimodal retrieval performance even without newly composed triplet datasets. Throughout the paper, we refer to this checkpoint as VISTA. Following UniIR [9], we adopt score-level fusion for fused embeddings when using CLIP-based models, referred to as CLIP-SF and TinyCLIP-SF for CLIP and TinyCLIP, respectively.

**Baselines** We compare the effectiveness of GCL with that of pretrained model and standard contrastive learning. For VISTA, we also compare ours with VISTA finetuned using the generated datasets composed of triplets (i.e., IT2I and T2IT) used in the original paper, which we denoted as CL+Triplet.

**Metrics** Following prior work [9, 27], we adopt the standard retrieval evaluation metric, Recall@K. Following UniIR, we set K=5 for the local setting, except for Fashion200K and FashionIQ, where we use K=10, and set K=50 for the global setting. We set K=1 for MMEB and K=1, 5, 10, and 50 for CoVR, following the previous work [27, 28]. Further details are provided in the Supplementary.

### 4.2 Quantitative Evaluation

Table 1 demonstrates that applying GCL consistently improves multimodal retrieval performance in the global setting of M-BEIR across various tasks and models. Notably, the performance improvement is particularly significant for tasks involving $q_{it}$ or $c_{it}$. Even without generating new data samples composed of $it$, GCL loss improves tasks related to $it$ with off-the-shelf image-caption paired datasets. While VISTA trained with generated datasets (i.e., IT2I and T2IT) shows the best performance for

---

[3]M-BEIR and CoVR datasets are under the MIT license, and MMEB is under the Apache-2.0 license.
[4]VISTA, CLIP, and TinyCLIP are all under the MIT license.

Table 1: Comparisons on global setting of M-BEIR using Recall@50. CL and GCL indicates standard contrastive learning and our generalized contrastive learning, respectively. Triplet and Pairwise refers to training with newly composed triplet dataset and original image-text paired dataset, respectively.

| Task | Dataset | VISTA [19] | | | | CLIP-SF [9] | | |
|---|---|---|---|---|---|---|---|---|
| | | Pretrained | CL +Triplet | CL +Pairwise | GCL (Ours) +Pairwise | Pretrained | CL +Pairwise | GCL (Ours) +Pairwise |
| 1. $q_t \rightarrow c_i$ | VisualNews [36] | 5.36 | 1.64 | 9.29 | 16.64 | 0.08 | 0.00 | 6.70 |
| | MSCOCO [37] | 2.72 | 5.60 | 14.42 | 38.85 | 0.00 | 0.00 | 3.25 |
| | Fashion200K [38] | 0.00 | 0.00 | 0.00 | 4.25 | 0.00 | 0.00 | 0.00 |
| 2. $q_t \rightarrow c_t$ | WebQA [39] | 97.07 | 96.90 | 96.86 | 96.25 | 60.29 | 88.55 | 60.24 |
| 3. $q_t \rightarrow (c_i, c_t)$ | EDIS [40] | 25.15 | 44.37 | 36.90 | 49.06 | 23.39 | 34.19 | 54.43 |
| | WebQA [39] | 14.22 | 80.88 | 31.74 | 64.00 | 19.87 | 68.42 | 40.62 |
| 4. $q_i \rightarrow c_t$ | VisualNews [36] | 1.35 | 0.08 | 1.18 | 4.71 | 0.00 | 0.00 | 2.48 |
| | MSCOCO [37] | 12.90 | 0.50 | 26.82 | 60.32 | 0.00 | 0.00 | 24.84 |
| | Fashion200K [38] | 0.02 | 0.00 | 0.00 | 0.72 | 0.00 | 0.00 | 0.16 |
| 5. $q_i \rightarrow c_i$ | NIGHTS [41] | 76.60 | 83.07 | 79.39 | 82.50 | 81.65 | 88.07 | 85.09 |
| 6. $(q_i, q_t) \rightarrow c_t$ | OVEN [42] | 5.06 | 1.78 | 3.10 | 8.72 | 0.00 | 0.00 | 3.63 |
| | InfoSeek [43] | 2.94 | 4.80 | 1.70 | 9.07 | 0.00 | 0.00 | 1.86 |
| 7. $(q_i, q_t) \rightarrow c_i$ | FashionIQ [44] | 6.66 | 16.41 | 6.10 | 10.88 | 11.61 | 0.00 | 4.25 |
| | CIRR [45] | 23.62 | 43.81 | 24.27 | 31.13 | 18.06 | 0.43 | 21.25 |
| 8. $(q_i, q_t) \rightarrow (c_i, c_t)$ | OVEN [42] | 34.31 | 9.67 | 32.83 | 32.92 | 11.04 | 0.58 | 19.47 |
| | InfoSeek [43] | 30.95 | 14.94 | 29.82 | 34.97 | 12.73 | 0.00 | 21.89 |
| | Avg. | 21.18 | 25.28 | 24.65 | **34.06** | 14.92 | 17.52 | **21.89** |

Table 2: Comparisons on MMEB dataset using Recall@1, following VLM2Vec [27]. Abbreviations as in Table 1.

| Task | Dataset | VISTA [19] | | | | CLIP-SF [9] | | |
|---|---|---|---|---|---|---|---|---|
| | | Pretrained | CL +Triplet | CL +Pairwise | GCL (Ours) +Pairwise | Pretrained | CL +Pairwise | GCL (Ours) +Pairwise |
| 1. $q_t \rightarrow c_i$ | VisDial [46] | 10.1 | 17.3 | 17.2 | 16.6 | 22.5 | 27.2 | 31.1 |
| | VisualNews [36] | 51.7 | 38.4 | 50.7 | 50.5 | 72.4 | 41.1 | 70.5 |
| | MSCOCO [37] | 32.8 | 44.8 | 46.8 | 48.7 | 54.9 | 60.7 | 61.5 |
| | Wiki-SS-NQ [47] | 16.3 | 12.4 | 14.7 | 16.7 | 50.7 | 34.1 | 46.5 |
| 2. $q_t \rightarrow c_{it}$ | WebQA [39] | 65.9 | 83.9 | 73.3 | 79.5 | 61.1 | 73.7 | 62.8 |
| | EDIS [40] | 78.0 | 64.6 | 78.2 | 78.5 | 79.2 | 45.4 | 85.4 |
| 3. $q_i \rightarrow c_t$ | VisualNews [36] | 54.6 | 25.7 | 52.7 | 54.2 | 1.5 | 0.2 | 10.9 |
| | MSCOCO [37] | 44.0 | 32.9 | 55.3 | 52.8 | 2.0 | 0.1 | 23.1 |
| 4. $q_i \rightarrow c_i$ | NIGHTS [41] | 64.7 | 64.1 | 65.7 | 65.4 | 60.1 | 9.1 | 66.4 |
| 5. $q_{it} \rightarrow c_i$ | CIRR [45] | 8.1 | 14.1 | 9.0 | 11.2 | 10.9 | 46 | 11.6 |
| | FashionIQ [44] | 3.3 | 9.0 | 3.1 | 7.7 | 9.9 | 16.5 | 6.2 |
| 6. $q_{it} \rightarrow c_{it}$ | OVEN [42] | 54.3 | 45.4 | 53.6 | 57.3 | 46.1 | 4.7 | 53.8 |
| | Avg. | 40.3 | 37.7 | 43.4 | **44.9** | 39.3 | 29.9 | **44.2** |

tasks $q_{it} \rightarrow c_i$ and $q_t \rightarrow c_{it}$, it shows limited performance gain or performance drop with other tasks. That is, finetuning retrieval models with generated samples under certain scenarios may show promising performance for the targeted scenarios, but they may fail to generalize to the tasks unseen during the finetuning phase. It also shows degraded performance on cross-modal tasks ($q_i \rightarrow c_t$ and $q_t \rightarrow c_i$) compared to the pretrained VISTA. We conjecture that further finetuning VISTA with composed sets targeting specific retrieval scenarios may degrade its original performance on cross-modal tasks due to forgetting its initial cross-modal alignment. We want to emphasize that our goal is to perform well across a wide range of tasks and datasets, not just to excel at a specific task or dataset.

Tables 2 and 3 compare the multimodal retrieval performances under the local setting. Again, finetuning retrieval models with GCL brings further performance improvements even under the local setting. Along with the global setting, we believe that performing well in the local setting is also important since we may need databases separately divided for each task depending on the use cases we pursue in the real-world applications. Although there may exist a slight performance drop in scenario-specific retrieval tasks (*e.g.*CIRR and FashionIQ), this can largely be attributed to the nature of the fine-tuning dataset used. The LCS-558K dataset is designed for general-purpose fine-tuning, which may not fully capture the nuances of domain-specific tasks. To achieve optimal performance in these specialized applications, we believe GCL serves as an effective initial training stage, and performance can be further improved through additional fine-tuning with task-specific data.

Table 4 demonstrates that applying GCL also improves the video retrieval performance. When deploying retrieval models in real-world scenarios, visual content may be stored in video formats

Table 3: Comparisons on local setting of M-BEIR. We report the results using Recall@5 for the local setting except using Recall@10 for Fashion200K and FashionIQ, following UniIR [9, 44]. Abbreviations as in Table 1.

| Task | Dataset | VISTA [19] | | | | CLIP-SF [9] | | |
|---|---|---|---|---|---|---|---|---|
| | | Pretrained | CL +Triplet | CL +Pairwise | **GCL (Ours) +Pairwise** | Pretrained | CL +Pairwise | **GCL (Ours) +Pairwise** |
| 1. $q_t \to c_i$ | VisualNews [36] | 16.04 | 10.01 | 15.78 | 15.42 | 44.34 | 20.97 | 36.71 |
| | MSCOCO [37] | 50.65 | 58.40 | 61.34 | 61.09 | 61.09 | 71.94 | 67.69 |
| | Fashion200K [38] | 9.31 | 8.03 | 9.83 | 9.54 | 6.57 | 8.84 | 7.04 |
| 2. $q_t \to c_t$ | WebQA [39] | 91.20 | 91.20 | 90.43 | 89.37 | 40.61 | 70.35 | 40.61 |
| 3. $q_t \to (c_i, c_t)$ | EDIS [40] | 36.69 | 40.98 | 35.76 | 45.88 | 43.29 | 34.56 | 48.97 |
| | WebQA [39] | 33.49 | 74.51 | 36.16 | 62.49 | 45.48 | 69.97 | 44.01 |
| 4. $q_i \to c_t$ | VisualNews [36] | 14.03 | 4.42 | 13.35 | 13.70 | 41.78 | 20.18 | 30.53 |
| | MSCOCO [37] | 61.66 | 60.44 | 71.98 | 72.56 | 79.00 | 85.78 | 79.04 |
| | Fashion200K [38] | 9.63 | 6.71 | 9.29 | 9.31 | 7.71 | 8.65 | 8.55 |
| 5. $q_i \to c_i$ | NIGHTS [41] | 26.32 | 26.32 | 28.21 | 28.35 | 26.13 | 30.94 | 30.99 |
| 6. $(q_i, q_t) \to c_t$ | OVEN [42] | 30.39 | 25.93 | 29.91 | 31.82 | 0.31 | 0.23 | 8.93 |
| | InfoSeek [43] | 29.87 | 23.16 | 28.47 | 34.26 | 0.29 | 0.00 | 6.78 |
| 7. $(q_i, q_t) \to c_i$ | FashionIQ [44] | 2.43 | 9.03 | 2.25 | 5.00 | 6.95 | 11.48 | 5.28 |
| | CIRR [45] | 10.60 | 21.82 | 11.34 | 14.27 | 13.19 | 37.84 | 15.85 |
| 8. $(q_i, q_t) \to (c_i, c_t)$ | OVEN [42] | 37.45 | 31.11 | 35.84 | 40.60 | 19.94 | 0.37 | 31.40 |
| | InfoSeek [43] | 23.08 | 28.34 | 23.94 | 35.32 | 19.40 | 0.13 | 24.28 |
| | Avg. | 30.18 | 32.53 | 31.49 | **35.56** | 28.51 | 29.51 | **30.42** |

Table 4: Comparisons on CoVR Benchmark. Following CoVR, we use the frame of middle index for the query video, while averaging 15 uniformly sampled frames for the target video. Abbreviations as in Table 1.

| Rank | VISTA [19] | | | CLIP-SF [9] | | |
|---|---|---|---|---|---|---|
| | Pretrained | CL +Pairwise | **GCL (Ours) +Pairwise** | Pretrained | CL +Pairwise | **GCL (Ours) +Pairwise** |
| R@1 | 31.22 | 33.76 | **37.52** | 37.32 | 19.68 | **37.60** |
| R@5 | 58.37 | 59.74 | **63.46** | 62.60 | 40.30 | **65.69** |
| R@10 | 68.15 | 69.52 | **72.81** | 71.99 | 50.67 | **75.78** |
| R@50 | 88.50 | 88.50 | **91.12** | 88.18 | 74.92 | **92.92** |

(*e.g.,* detecting unexpected actions in CCTV), making video retrieval an important task due to its practicality. Consistent performance improvements in multimodal retrieval tasks even including video retrieval demonstrates that GCL is a robust and versatile approach for enhancing retrieval models across diverse scenarios.

## 5 Further Analysis

### 5.1 Ablation Studies

Table 5 compares GCL with the intra-modality separation loss proposed in AlignCLIP [11] while dissecting the contributions of the individual loss components of GCL. $\mathcal{L}_{a2b}$ indicates the loss function of GCL using $a$ as the query modality and $b$ as the target modality from a given positive pair. Regarding intra-modality separation loss, we added the loss term in addition to standard contrastive learning during training. For the ablation study of GCL, we excluded each of the following key loss functions: 1) cross-modal alignment terms ($\mathcal{L}_{i2t}$ and $\mathcal{L}_{t2i}$), 2) *it*-candidate learning terms ($\mathcal{L}_{i2it}$ and $\mathcal{L}_{t2it}$), and 3) *it*-query learning terms ($\mathcal{L}_{it2i}$ and $\mathcal{L}_{it2i}$). For the comparisons, we use the global setting of M-BEIR. Results on the local setting of M-BEIR and performance variance of multiple runs are included in our Supplementary.

As shown, adding the intra-modality separation loss indeed improves the multimodal retrieval performance compared to training with standard contrastive learning. However, we observe that the performance gain is limited for tasks involving retrieval with identical modalities (*e.g.*, $q_i \to c_i$ and $q_t \to c_t$) or queries with *it* modality (*e.g.*, $q_{it} \to c_i$ and $q_{it} \to c_t$) compared to our GCL loss. This indicates that intra-modality separation loss mitigates the modality gap but it fails to consider diverse multimodal retrieval scenarios, which are effectively addressed by GCL.

Regarding the ablation study, we observe a performance drop on the task that each module of GCL loss is responsible for. To be more specific, by excluding $\mathcal{L}_{i2t}$ and $\mathcal{L}_{t2i}$, the performance on cross-modal tasks are degraded significantly. Also, the performances on tasks of $q_t \to c_{it}$ are degraded after excluding $\mathcal{L}_{i2it}$ and $\mathcal{L}_{t2it}$. We want to emphasize that we did not perform an extensive hyperparameter search for finding the optimal weighing values for each loss function in GCL. While

Table 5: Ablation studies on loss functions and comparisons with intra-modality separation loss [11] using global setting of M-BEIR.

| Task | Dataset | CL | Intra-modality Separation [11] | GCL w/o $\mathcal{L}_{i2t}, \mathcal{L}_{t2i}$ | GCL w/o $\mathcal{L}_{i2it}, \mathcal{L}_{t2it}$ | GCL w/o $\mathcal{L}_{it2i}, \mathcal{L}_{it2t}$ | **GCL** |
|---|---|---|---|---|---|---|---|
| 1. $q_t \rightarrow c_i$ | VisualNews [36] | 9.29 | 14.36 | 2.91 | 18.26 | 17.07 | 16.64 |
|  | MSCOCO [37] | 14.42 | 36.67 | 9.77 | 39.43 | 38.99 | 38.85 |
|  | Fashion200K [38] | 0.00 | 3.84 | 0.41 | 3.90 | 4.54 | 4.25 |
| 2. $q_t \rightarrow c_t$ | WebQA [39] | 96.86 | 96.17 | 97.68 | 96.13 | 96.21 | 96.25 |
| 3. $q_t \rightarrow (c_i, c_t)$ | EDIS [40] | 36.90 | 49.74 | 45.23 | 37.15 | 49.18 | 49.06 |
|  | WebQA [39] | 31.74 | 47.59 | 69.53 | 52.93 | 61.65 | 64.00 |
| 4. $q_i \rightarrow c_t$ | VisualNews [36] | 1.18 | 2.78 | 0.59 | 5.50 | 4.63 | 4.71 |
|  | MSCOCO [37] | 26.82 | 48.64 | 13.4 | 63.06 | 58.60 | 60.32 |
|  | Fashion200K [38] | 0.00 | 0.61 | 0.08 | 0.76 | 0.63 | 0.72 |
| 5. $q_i \rightarrow c_i$ | NIGHTS [41] | 79.39 | 78.02 | 79.15 | 83.21 | 82.83 | 82.50 |
| 6. $(q_i, q_t) \rightarrow c_t$ | OVEN [42] | 3.10 | 5.23 | 6.92 | 9.77 | 7.95 | 8.72 |
|  | InfoSeek [43] | 1.70 | 3.72 | 9.10 | 10.96 | 7.79 | 9.07 |
| 7. $(q_i, q_t) \rightarrow c_i$ | FashionIQ [44] | 6.10 | 6.33 | 9.48 | 11.76 | 10.61 | 10.88 |
|  | CIRR [45] | 24.27 | 23.57 | 32.52 | 31.82 | 30.50 | 31.13 |
| 8. $(q_i, q_t) \rightarrow (c_i, c_t)$ | OVEN [42] | 32.83 | 35.74 | 36.75 | 29.20 | 31.22 | 32.92 |
|  | InfoSeek [43] | 29.82 | 34.26 | 40.26 | 33.31 | 32.27 | 34.97 |
|  | Avg. | 24.65 | 30.45 | 28.36 | 32.95 | 33.42 | **34.06** |

Table 6: Performance improvements on M-BEIR under global setting with TinyCLIP.

| Metric | VISTA | CLIP-SF | TinyCLIP-SF | TinyCLIP-SF + GCL |
|---|---|---|---|---|
| Model Params. | 196M | 427M | 120M | 120M |
| Avg. Inference (ms) | 26.06 | 21.58 | 14.67 | 14.67 |
| M-BEIR | 21.18 | 14.92 | 17.36 | **22.71** |

this may improve performance on certain tasks, it may accompany performance drops on other tasks. Since the main goal of this paper is to design a loss function that generally works well for diverse retrieval scenarios, we simply added the loss functions with identical weights. Depending on the use cases, each loss function in GCL may be weighed differently.

## 5.2 Empowering Lightweight Models

Deploying lightweight models for retrieval is essential for enabling fast and efficient inference in real-time or resource-constrained environments, such as mobile or edge devices (*e.g.,* retrieving personal data on mobile phones). Table 6 illustrates that applying GCL improves the retrieval performance of TinyCLIP compared to pretrained retrieval models, including VISTA and CLIP-SF, despite having fewer parameters and a lower average inference speed (ms). This result indicates that fine-tuning a small and lightweight retrieval model with GCL is a viable solution for improving retrieval performance.

## 5.3 Ranks of Ground Truth Candidates

Figure 4 compares the ranks of ground truth candidates between VISTA and VISTA trained with GCL. The x-axis denotes the ranks, and the y-axis indicates its frequency. For this analysis, we use $q_t$ from MSCOCO and a candidate pool composed of $c_t$ and $c_i$ from MSCOCO, where the task is to retrieve the ground truth $c_i$ given $q_t$. The numbers of queries and candidates are 2.5K and 29K, respectively. We visualize only candidates ranked within the top 10K.

Our findings show that when VISTA is trained with GCL, most ground truth candidates achieve high ranks, with the majority ranked within the top 500. In contrast, VISTA without GCL exhibits a non-trivial number of ground truth candidates that are ranked significantly lower. This highlights the challenge of retrieving ground truth $c_t$ from a shared database of mixed modalities when the modality gap is not effectively reduced. GCL successfully mitigated the modality gap, as demonstrated by the high ranks of ground truth candidates.

## 5.4 Cosine Similarity with Candidates

**Ground truth candidates** Figure 5 (a) visualizes the cosine similarity between the embeddings of queries and their corresponding ground truth candidates. For this analysis, we selected one dataset from each task in M-BEIR. As shown, applying GCL to VISTA consistently increases the cosine similarities across diverse tasks and datasets, indicating improved alignment between query and ground truth representations. By improving the representation space, GCL ensures that relevant

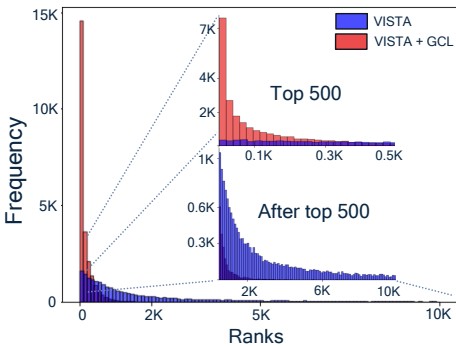

Figure 4: Rankings of ground truth candidates. The x-axis and y-axis indicate the ranks and the frequency of ranks, respectively. We use the task of $q_t \rightarrow c_i$ on the MSCOCO dataset, with a candidate pool composed of $c_i$ and $c_t$ from MSCOCO.

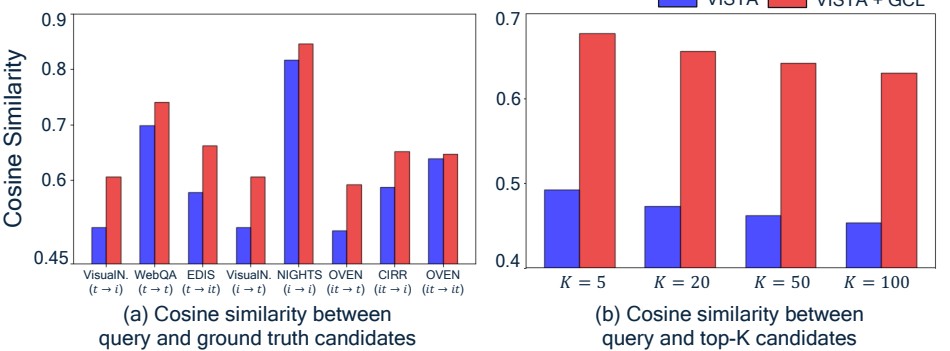

(a) Cosine similarity between query and ground truth candidates

(b) Cosine similarity between query and top-K candidates

Figure 5: (a) Cosine similarity between query and ground truth candidates. X-axis and y-axis indicates the dataset and cosine similarity, respectively. VisualN. refers to VisualNews. (b) Cosine similarity between queries and top-ranked candidates. We use MSCOCO for the task of $q_i \rightarrow c_t$.

multimodal pairs - whether text, image, or fused - are positioned closer in the embedding space, leading to more accurate retrieval.

**Top-ranked candidates** Figure 5 (b) further examines the cosine similarity between queries and their top-ranked retrieved candidates to evaluate the retrieval consistency. Using MSCOCO ($q_i \rightarrow c_t$), we analyze how cosine similarity trends change across different ranks. As shown, VISTA exhibits a significant drop in similarity with lower ranks, suggesting that lower-ranked candidates are less semantically relevant. In contrast, applying GCL helps maintain high cosine similarity even with lower ranks, demonstrating that relevant candidates are still retrieved even with lower ranks.

This stability across ranks highlights the ability of GCL to build a unified representation space more effectively, ensuring that even when the top candidate is not a perfect match, subsequent retrieved items remain semantically meaningful. Such improvements are crucial for real-world multimodal retrieval applications where retrieving a set of relevant candidates would be a viable solution, rather than retrieving the single best match.

## 6 Conclusion

In this paper, we introduced Generalized Contrastive Learning (GCL), a *simple yet effective* loss function designed to enhance multimodal retrieval performance without the need for generating triplet datasets simulating certain retrieval scenarios. By integrating text, image, and fused text-image embeddings into the contrastive learning framework, GCL mitigates the modality gap and improves multimodal retrieval performance across diverse tasks and datasets. Although not discussed in this work, one promising future work direction is integrating GCL with multimodal large language models (MLLMs) to further enhance retrieval capabilities in generative and reasoning-based tasks [23, 21, 22, 48, 49, 50]. As MLLMs continue to advance, utilizing retrieved information to generate responses would be promising, especially with databases containing mixed-modalities. By eliminating the need for labor-intensive dataset curation while improving retrieval across arbitrary modality combinations, GCL presents a scalable and effective solution for multimodal retrieval. We hope this work paves the way for future research in leveraging contrastive learning for more generalizable and robust multimodal retrieval.

**Acknowledgement** We would like to thank Qualcomm AI Research team for their valuable discussions.

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
