# OpenReview forum: "Generalized Contrastive Learning for Universal Multimodal Retrieval"
_NeurIPS.cc/2025/Conference — NeurIPS 2025 poster_

### Official Review · Reviewer_h6VU · 2025-06-29

**Clarity:** 3
**Significance:** 2
**Originality:** 3
**Rating:** 4
**Confidence:** 4

**Summary:**

The paper introduces Generalized Contrastive Learning (GCL), a novel loss function designed to improve multimodal retrieval performance by leveraging existing image-caption paired datasets without requiring costly curated triplet datasets. GCL integrates text, image, and fused text-image embeddings into a unified representation space through contrastive learning, addressing the modality gap and enabling retrieval across diverse modality combinations. The method is evaluated on benchmarks like M-BEIR, MMEB, and CoVR, showing consistent improvements across models such as VISTA, CLIP, and TinyCLIP.

**Questions:**

1. Clarify the process of (q_i, t_t) -> c_i. Could you elaborate on how the mapping from (q_i, t_t) to c_i is performed?

2. Distinguish between (q_i, q_t) -> (c_i, c_t) and q_it -> c_it. What are the key differences between these two tasks?

3. Consider using wikipedia articles or other easy-obtained datasets for experiments. Since multimodal wikipedia articles are readily accessible, why were they not used in the experiments? Their inclusion could provide additional validation and broader applicability.

**Ethical Concerns:**

["NO or VERY MINOR ethics concerns only"]

**Final Justification:**

The concerns have been addressed. And I keep change my rating.

**Limitations:**

Yes.

**Paper Formatting Concerns:**

no such concerns.

**Quality:**

3

**Strengths And Weaknesses:**

Strengths:

- The proposed loss function is novel and demonstrates improved performance over existing methods.

- The paper provides extensive experimental results, facilitating a deeper understanding and thorough analysis of the method.

Weaknesses:

- The core idea is validated only on existing datasets, which were not specifically designed for this method. A specially constructed dataset containing challenging samples (e.g., q_it -> c_it) would further strengthen the contribution.

- The loss function defined in Eq. (2) and illustrated in Fig. 3 appears relatively easy to optimize, as the positive pairs are derived solely from different modality combinations of the SAME sample, potentially limiting its robustness.

---

> ### Author Rebuttal · Authors · 2025-07-30
>
> We are grateful for the reviewer’s recognition of the novelty of our proposed loss function and the comprehensiveness of our experimental results. This feedback aligns with Reviewer g4sT’s acknowledgments of our method’s effectiveness, simplicity, and empirical depth. Regarding the concerns and questions raised, we have systematically addressed each point below, and all corresponding modifications will be incorporated into the final manuscript.
>
> **[W1, Q2, Q3] Clarification on Evaluation**
>
> We appreciate the reviewer’s suggestion regarding the use of specially constructed datasets that are both challenging and well aligned with our proposed method (e.g., including the $q_{it} \rightarrow c_{it}$ task). However, we would like to clarify that our experiments already include evaluations on $q_{it} \rightarrow c_{it}$ retrieval tasks using existing datasets. Specifically, both OVEN and InfoSeek include tasks of this nature, as shown in Table 1 - Task 8: ($q_{i}, q_{t}$) $\rightarrow$ ($c_{i}, c_{t}$). We confirm that GCL shows strong improvements in these settings.
>
> Also, although the tasks ($q_{i}, q_{t}$) $\rightarrow$ ($c_{i}, c_{t}$) and $q_{it} \rightarrow c_{it}$ represent the same retrieval task, they were denoted differently in Tables 1 and 2. Recognizing this potential ambiguity, we will unify the notation as $q_{it} \rightarrow c_{it}$ for clarity in our camera-ready version.
>
> Regarding the inclusion of Wikipedia in our evaluation, we clarify that our experimental setup already includes several datasets derived from Wikipedia articles and images. Details of the test sets using Wikipedia-derived samples are as follows:
> - WebQA: Uses Wikipedia pages (text and images) as retrieval candidates for answering multimodal questions.
>
> - OVEN: Provides visual recognition questions with Wikipedia images and corresponding summaries as reference targets.
>
> - InfoSeek: Evaluates VQA and retrieval on web-based multimodal content, including Wikipedia-derived samples.
>
> - Wiki-SS-NQ: A retrieval-based VQA dataset using Wikipedia page screenshots, offering richer context than plain text.
>
> These demonstrate that the textual and visual content of Wikipedia is already well represented in our test sets. Additionally, the three benchmarks used in our work, M-BEIR (16 subtests), MMEB (12 subtests), and CoVR (1 subtest), covers 29 subtests in total. By showing consistent performance across these tests, we believe that we demonstrated the generalizability and broad applicability of GCL.
>
> **[W2] Optimization Simplicity of GCL Loss**
>
> While positive pairs in GCL originate from different modalities of the same semantic instance, this does not make the optimization trivial. In fact, the optimization remains non-trivial due to the inherent modality gap. As illustrated in the PCA visualization of Figure 2 in our main paper, even semantically aligned content (e.g., an image and its corresponding caption) often lies in distant regions of the representation space. The image-text fused modality introduces an additional alignment challenge, as it requires learning a unified space that accommodates all three modalities. If the GCL loss was trivial to optimize, we would expect these semantically related samples across modalities to be closely clustered in the representation space, which is clearly not the case. These observations underscore that GCL performs meaningful optimization across heterogeneous modalities and is essential for bridging the modality gap in multimodal retrieval.
>
> **[Q1] Clarification on $q_{i,t}$ $\rightarrow$ $c_{i}$ retrieval setting**
>
> The notation $q_{i,t}$ $\rightarrow$ $c_{i}$ refers to a compositional image retrieval task. In this setup, the query is composed of an image and a modifying textual phrase, and the goal is to retrieve an image that reflects the modified concept. For example, if the query consists of an image of a red dress and the text "is white with dots and a black belt," the target image $c_{i}$ would be an image of a "white dress with dots and a black belt." Thus, the task requires compositional reasoning to interpret the modification and apply it to the visual concept. We will describe the details of each task in our camera-ready version.

---

> > ### Comment · Reviewer_h6VU · 2025-08-05
> > **Response**
> >
> > Many thanks for addressing all my points so completely. No further questions for now.

---

> ### Comment · Reviewer_kEbb · 2025-08-07
>
> Thanks for the response, I'll raise my score by 1 accordingly

---

### Official Review · Reviewer_kEbb · 2025-07-02

**Clarity:** 2
**Significance:** 2
**Originality:** 2
**Rating:** 3
**Confidence:** 4

**Summary:**

This paper proposes Generalized Contrastive Learning (GCL), a loss function for multimodal retrieval that integrates text, image, and fused image-text embeddings within a single mini-batch for contrastive learning. The method aims to address the modality gap problem in retrieving keys with mixed modalities (e.g., Wikipedia pages containing both images and text) without requiring expensive triplet dataset curation. The authors demonstrate improvements on M-BEIR, MMEB, and CoVR benchmarks across VISTA, CLIP, and TinyCLIP models.

**Questions:**

Three fundamental questions require clarification before this work can be properly evaluated:

Why does the method fail on certain tasks despite using the same underlying principle? The substantial performance drops on tasks like WebQA (qt→ct) in Table 1 suggest the approach may not be as robust as claimed. What are the failure modes, and can they be predicted a priori?

How does computational cost scale with the number of modalities and batch size? The GCL loss requires computing similarities across all modality combinations within each batch, potentially creating a quadratic scaling issue. The authors provide no analysis of training time or memory requirements compared to standard contrastive learning.

What is the theoretical basis for equal weighting of all loss components? Real-world datasets have varying amounts of data across modalities and differing task difficulties. The assumption that all six modality pairs should contribute equally to the loss seems unjustified and warrants either theoretical support or empirical validation through extensive hyperparameter search.

**Ethical Concerns:**

["NO or VERY MINOR ethics concerns only"]

**Final Justification:**

I've read the rebuttal and decided to raise my score by 1.

**Limitations:**

yes

**Quality:**

2

**Strengths And Weaknesses:**

**Strengths:**
The central premise of the work is reasonable - current cross-modal retrieval models struggle with fused modality combinations, and existing approaches that generate specialized triplet datasets are computationally expensive and fail to generalize.

**Weaknesses:**
The method is relatively simple, but whether it is effective needs more experimental validation to support the claims and it is hard to find potential insight behind the method. This makes the work appear to be incremental work based on VISTA and lacks insights and experimental depth equivalent to VISTA.

My main concern is whether the NeurIPS community truly needs such incremental image-text matching work, even if the experiments are very comprehensive. This type of work might be more suitable for communities like MM.

The reproducibility is insufficient - there are surprisingly 0 pages of appendix, which is unacceptable for the NeurIPS community that requires highly reproducible work. For future three top-tier AI conference submissions, it is recommended to place the appendix after the main text.

From Table 1, we can see that this method does not always work. For example, in the WebQA qt→ct task, GCL's performance is worse than the baseline. However, the authors did not add bold or underline to highlight this, attempting to hide this point.

**Minor revision suggestions:**
1. The text shape overlap in Figure 1 can be optimized, such as for the `fused text embedding`.
2. The related work analysis is not comprehensive enough. It fails to serve the purpose of discussing the development trajectory, with many works crammed into single sentences.
3. Figure 4 could save space by using half the space with such a figure layout.

---

> ### Author Rebuttal · Authors · 2025-07-30
>
> We appreciate the reviewer’s acknowledgment of the core motivation behind our work. This aligns with Reviewer PYSB’s appreciation of our problem framing and Reviewer g4sT’s recognition of the method’s effectiveness without relying on dense multimodal pairings. We have clarified points of misunderstanding and concerns and will incorporate all necessary changes into the final camera-ready manuscript.
>
> **[W1] Clarification on Novelty and Experimental Depth**
>
> We appreciate the reviewer’s concern and agree that GCL is conceptually simple. However, we believe this simplicity is a strength of GCL because it is lightweight, broadly applicable, and easy to integrate into existing frameworks. Its core novelty lies in generalizing contrastive learning to a unified multimodal retrieval setting without requiring complex data curation (e.g., triplet synthesis), which was necessary in previous multimodal retrieval studies, including VISTA, as the reviewer noted. Furthermore, our approach is grounded in the observation that the modality gap remains significant, especially when the fused $it$ modality is incorporated, as shown in Figure 2 of the main paper.
>
> Below are the detailed uniqueness and experimental depth of our work compared to VISTA:
> - Training with off-the-shelf image-caption datasets: While VISTA requires constructing task-specific datasets, GCL operates directly on standard image-caption datasets. This eliminates the need for labor-intensive dataset curation and demonstrates broader applicability.
> - Diverse Model Evaluations: GCL is applied not only to VISTA, but also to CLIP and TinyCLIP, demonstrating its architecture-agnostic nature. This highlights that GCL is not merely a VISTA add-on, but a general-purpose, plug-and-play loss function.
> - Broad Benchmarks: We evaluate across M-BEIR (16 subtests), MMEB (12 subtests), and CoVR (1 subtest), including the video retrieval task, which was not addressed by VISTA. We demonstrate the effectiveness of GCL across 29 subtests in total, while VISTA evaluated only 5 subtests.
> - Global and Local Retrieval: GCL consistently performs well across both global and local retrieval settings, while VISTA conducted experiments only under the local retrieval setting.
>
> While we referred to VISTA multiple times, this was to provide fair and direct comparisons with a recent state-of-the-art method that relies heavily on curated datasets. We will further clarify these distinctions in the revised version to better highlight the contributions and uniqueness of GCL along with the exhaustive experiments conducted in our work.
>
> **[W2] Suitability of our work for NeurIPS**
>
> First, we want to clarify that GCL is not limited to traditional image-text matching. Our main goal is to improve the representation space by reducing the modality gap, thereby enhancing universal multimodal retrieval performance overall. To this end, we believe that our work fits within the "General Machine Learning" category at NeurIPS.
>
> Second, GCL demonstrates that a single objective function trained only on existing image-caption pairs can generalize to multiple modalities and tasks. This design aligns with the NeurIPS community’s interest in efficient, scalable, and generalizable machine learning frameworks. We believe GCL is well-aligned with these priorities.
>
> Lastly, while not addressing the identical problem setup, we note that research focused on improving retrieval or embedding spaces has been well represented in recent NeurIPS [1,2,3], ICLR [4], and ICML [5] proceedings. Building on this precedent, we believe our work continues this line of inquiry while making a distinct contribution focused on multimodal retrieval through its broad applicability and empirical thoroughness. We respectfully suggest that GCL is well suited for and valuable to the NeurIPS audience.
>
> [1] Self-Retrieval: End-to-End Information Retrieval with One Large Language Model (NeurIPS 2024)
>
> [2] Multi-Lingual Acquisition on Multimodal Pre-training for Cross-modal Retrieval (NeurIPS 2022)
>
> [3] Large-Scale Retrieval for Reinforcement Learning (NeurIPS 2022)
>
> [4] MM‑Embed: Universal Multimodal Retrieval with Multimodal LLMs (ICLR 2025)
>
> [5] Contradiction Retrieval via Contrastive Learning with Sparsity (ICML 2025)
>
>
> **[W3] Missing Appendix and Reproducibility**
>
> We would like to clarify that we had already submitted an 8-page comprehensive appendix as a separate file, following the NeurIPS guidelines. Our supplementary material includes pseudocode for GCL, detailed implementation settings, additional results and analyses, and compute resources. In fact, we already indicated in our initial submission that certain content is included in the supplementary (see lines 196 and 231, as well as Checklist items 4, 6, 8, and 10 in the main paper). Additionally, in supplementary Section B, we noted that our code is currently under legal review and will be released upon approval. Since the NeurIPS guidelines do not penalize authors for submitting the appendix separately from the main text, we believe this review comment stems from a simple misunderstanding of our submitted materials.
>
> **[W4, Q1] Performance Drop in WebQA and Formatting**
>
> First, we acknowledge the slight performance drop observed in WebQA ($q_t \rightarrow c_t$) in Table 1. Our primary objective was to improve performance across a broad range of multimodal retrieval tasks without relying on dataset construction tailored to specific retrieval objectives. In this context, we believe the minor drop in WebQA (from 97.07 to 96.25) is acceptable, especially considering that the task may already be saturated and less receptive to gains from learning other tasks.
>
> Second, we would like to clarify that we did not intend to hide this performance drop. The reviewer noted that we did not bold or underline the degraded score, implying an attempt to hide it. However, the standard practice is to highlight only the best-performing results per task, not to indicate drops. We have transparently reported all sub-task results, including those with degraded performance, and will explicitly discuss such cases in the limitation section of the final camera-ready version.
>
> **[Minor] Presentation (Figures and Related Work)**
>
> We thank the reviewer for the helpful suggestions. We will revise Figure 1 to improve readability by reducing label overlap, especially for fused text embeddings, and redesign Figure 4 for more efficient use of space. We also agree that the related work section could better reflect the development trajectory and will expand it with a more structured discussion, moving less critical references to the appendix if needed.
>
> **[Q2] Computational Costs**
>
> Importantly, GCL introduces no additional memory or latency overhead during inference, as it modifies only the training loss function and leaves the model architecture unchanged. This ensures that the deployed model remains as efficient as those trained with standard contrastive learning. We will include a detailed analysis and supporting data in the final camera-ready version.
>
> During training, GCL does incur some additional computational and memory costs due to the need to compute similarities across a broader set of modality combinations. While standard contrastive learning typically computes similarities between two modalities (e.g., image-text), GCL extends this to include four additional modality pairs, such as fused combinations. Furthermore, GCL introduces an additional forward pass for the fused image-text ($it$) modality, which contributes to additional time overhead.
>
> To quantify this computational overhead, we compared peak VRAM usage and per-iteration training time between standard contrastive learning and our proposed GCL. This analysis was conducted using a batch size of 128 on a single NVIDIA H100 GPU, averaged over 100 iterations with the VISTA model. The results show a modest increase in both VRAM usage and training time per iteration. However, we believe this overhead is well justified given the performance improvements GCL achieves across a wide range of tasks and benchmarks.
>
> | Method | Peak VRAM | Time     |
> |---|---|---|
> | Standard  | 24.45 GB  | 0.3325 s/iter |
> | GCL    | 45.78 GB  | 0.6074 s/iter |
>
> **[Q3] Equal Loss Weighting Justification**
>
> As described in the original manuscript (lines 241–246), we chose to apply equal weights to each loss term to avoid overfitting to specific modality pairs or retrieval tasks. Our objective was not to tune the model for the best performance on individual datasets but rather to demonstrate the general applicability and robustness of the proposed GCL loss function.
>
> To further validate this choice, we conducted an ablation study on loss weighting, varying the loss coefficients for three distinct tuples of loss terms: {($\mathcal{L}{\scriptstyle{t2i}}$, $\mathcal{L}{\scriptstyle{i2t}}$), ($\mathcal{L}{\scriptstyle{t2it}}$, $\mathcal{L}{\scriptstyle{i2it}}$), ($\mathcal{L}{\scriptstyle{it2i}}$, $\mathcal{L}{\scriptstyle{it2t}}$)}. For each tuple, we varied the coefficient values over {0, 0.1, 1, 10}, while keeping all other loss terms fixed at 1. Due to the space limit, we could only report the average scores of each experiment. If requested, we will also show the results of each sub test during the discussion period.
>
> |       | {1,1,1} | {0,1,1} | {0.1,1,1} | {10,1,1} | {1,0,1} | {1,0.1,1} | {1,10,1} | {1,1,0} | {1,1,0.1} | {1,1,10} |
> |---|---|---|-----|-----|------|----|----|-----|----|---|
> | **Average** | 34.06   | 28.36   | 34.59     | 33.02    | 32.95   | 33.65      | 34.02     | 33.42    | 33.39      | 34.50     |
>
> While certain weighting configurations yielded slight improvements, we found that equal weighting provides consistently strong results across different tasks and datasets. We will include the full results of this hyperparameter search in the camera-ready version.

---

### Official Review · Reviewer_PYSB · 2025-07-06

**Clarity:** 2
**Significance:** 2
**Originality:** 2
**Rating:** 3
**Confidence:** 4

**Summary:**

This paper proposes Generalized Contrastive Learning (GCL), a loss function designed to improve universal multimodal retrieval. The method targets a key weakness in standard cross-modal models like CLIP: their poor performance when retrieving composite items containing both images and text (e.g., a Wikipedia page). Instead of the costly approach of curating new triplet-based datasets for specific retrieval scenarios (e.g., text query to image-text key), GCL works by simply extending standard contrastive learning to a three-modality setting. Using existing image-caption datasets, it generates embeddings for the image (e_i), the text (e_t), and a fused image-text representation (e_it). It then applies a contrastive loss across all three modalities within a mini-batch to learn a more unified representation space. The authors demonstrate consistent performance gains on several benchmarks (M-BEIR, MMEB, CoVR) when applying GCL to off-the-shelf models like VISTA and CLIP.

**Questions:**

Please check the above for details.

**Ethical Concerns:**

["NO or VERY MINOR ethics concerns only"]

**Final Justification:**

The reviewer appreciates the authors' rebuttal.

The reviewer meant to say [12] in the original review but the reviewer thinks it is reasonable to expect the the settings where one of the first papers identified modality gap of CLIP embedding would be carefully revisited in a following-up work that claims mitigating the gap. Specifically, Section 5 in [12], including more serious study on relationship between modality gap measure and downstream tasks, as well as the performance in fairness setup.

Given this as well as the incoherent performance on some of the tasks without further justification in the settings like fine-tuning which the author mentioned, the reviewer cannot recommend accept. But the reviewer believe the manuscript is potentially important for the community once these issues are fully resolved.

**Limitations:**

Need more discussion on the possible drawback of the method.

**Quality:**

2

**Strengths And Weaknesses:**

Pros:
1. The targeted problem is important.
2. The proposed idea is interesting and straightforward.
3. The paper is overall easy to follow and well presented.

Cons:
1. The authors claimed that this reduces the modality gap in Figure 2, which is attractive. However, there is no following up study on how the minimized modality gap translates to the improvement in downstream performance. Specifically, the datasets and evaluations used in [1] to demonstrate the harm of modality gap are not revisited to truly demonstrate the benefit of smaller modality gap.

2. The results are not very convincing.
- a. In Table 1, the baseline CLIP+Pairwise under CLIP-SF has much worse performance than the pretrained model, which is quite inconsistent with the the trend in Table 2 and Table 3.
- b. In Table 2 and 3, the proposed method actually harms the performance on datasets like CIRR and FashionIQ, which indicates that the proposed is not that general and it might not be in an ideal formulation.

---

> ### Author Rebuttal · Authors · 2025-07-30
>
> We deeply appreciate the reviewer’s recognition of the importance of the addressed problem, the simplicity and clarity of our method, and the overall presentation of the paper. We are especially encouraged by the acknowledgment that the proposed reduction in modality gap is an attractive property. We would like to note that Reviewer g4sT similarly praised the benefits of aligning different modality combinations, as demonstrated in our empirical design (e.g., Figure 2). Regarding the concerns raised, we have carefully reviewed and addressed each point below, and all necessary changes will be faithfully incorporated into the final camera-ready submission.
>
> **[W1] Relationship with Modality Gap and Downstream Task Performance**
>
> We appreciate the reviewer’s thoughtful observation about the modality gap in Figure 2 and the request for stronger evidence connecting it to downstream performance. We want to emphasize that one of our main goals is to retrieve the ground truth candidate from a database composed of samples with different modalities, which is referred to as the "global setting" in our work. In this context, embeddings from different modalities must reside in a unified representation space, making it essential to minimize the modality gap.
>
> To further demonstrate our point, we analyzed the modalities of the top-100 retrieved candidates in the global setting of M-BEIR using CLIP-SF and report the ratio of each modality. We observe that the modalities of the top-100 candidates are mostly identical to the modality of the query, rather than the candidate. For example, when the query modality is text, the majority of the top-100 retrieved candidates are also text, even when corresponding paired images are available. This suggests that the modality gap inherent in the representation space of CLIP biases retrieval toward the same modality as the query, regardless of semantic alignment, which results in degraded performance in downstream tasks. We will include this analysis in the camera-ready version.
>
> | Query → Cand.        | Dataset | Image (\%) | Text (\%) | Image + Text (\%) |
> |--------------------------|----------------|-----------|----------|------------------|
> | $q_t \rightarrow c_i$    | VisualNews     | 0.00      | 97.52    | 2.48             |
> |                          | MSCOCO         | 0.00      | 99.97    | 0.03             |
> |                          | Fashion200K    | 0.00      | 100.00   | 0.00             |
> | $q_t \rightarrow c_t$    | WebQA          | 0.00      | 99.84    | 0.16             |
> | $q_t \rightarrow c_{it}$ | EDIS           | 0.00      | 95.04    | 4.96             |
> |                          | WebQA          | 0.00      | 96.73    | 3.27             |
> | $q_i \rightarrow c_t$    | VisualNews     | 94.89     | 0.00     | 5.11             |
> |                          | MSCOCO         | 95.91     | 0.00     | 4.09             |
> |                          | Fashion200K    | 100.00    | 0.00     | 0.00             |
> | $q_i \rightarrow c_i$    | NIGHTS         | 99.03     | 0.00     | 0.97             |
> | $q_{it} \rightarrow c_t$ | OVEN           | 0.14      | 0.88     | 98.98            |
> |                          | InfoSeek       | 0.37      | 0.20     | 99.43            |
> | $q_{it} \rightarrow c_i$ | FashionIQ      | 63.54     | 0.33     | 36.12            |
> |                          | CIRR           | 32.09     | 0.09     | 67.82            |
> | $q_{it} \rightarrow c_{it}$ | OVEN        | 0.06      | 0.64     | 99.31            |
> |                          | InfoSeek       | 0.25      | 0.14     | 99.61            |
>
> Additionally, this motivation aligns with recent work such as AlignCLIP [11], which demonstrates that reducing the modality gap leads to improved cross-modal retrieval performance. While AlignCLIP focuses on the cross-modal retrieval setting and does not address retrieval across arbitrary combinations of modalities (e.g., involving the $it$ modality), our work addresses such multimodal retrieval settings. We further show that minimizing the modality gap using GCL leads to strong performance in these richer multimodal retrieval settings. As shown in Table 5 of the original manuscript, GCL outperforms the intra-modality separation loss proposed in AlignCLIP under diverse multimodal retrieval settings. This suggests that GCL offers a more effective strategy for unifying representations across multiple modalities.
>
> We will clarify this relationship between reducing the modality gap and improving multimodal retrieval performance in our camera-ready version. Additionally, since the reviewer did not specify the paper [1] in the initial review comment, we could not identify the datasets and evaluations used in [1]. However, we want to emphasize that our evaluation includes three comprehensive benchmarks, M-BEIR (16 subtests), MMEB (12 subtests), and CoVR (1 subtest), covering a total of 29 subtests. We believe this provides strong evidence of GCL’s generalizability and broad applicability to multimodal retrieval.
>
> [11] Mitigate the Gap: Improving Cross-Modal Alignment in CLIP (ICLR 2025)
>
> **[W2] Clarification on Results**
>
> We thank the reviewer for carefully examining the results and raising two important points.
>
> (a) Regarding the inconsistent performance of CLIP + Pairwise:
>
> The difference in performance trends between Table 1 (global setting) and Tables 2 and 3 (local setting) reflects a fundamental difference in retrieval context. CLIP-SF fine-tuned with pairwise data generally performs well in local settings, where candidates are retrieved only from the same modality and task, as shown in Tables 2 and 3. However, Table 1 shows that the representation space of CLIP is insufficient for global retrieval, where queries and candidates can come from arbitrary modality combinations. As further shown in the table above, which reports the modality ratios of the top-100 candidates, CLIP-SF in the global setting tends to retrieve candidates that share the same modality as the query, making it difficult to retrieve the ground truth candidates. We initially discussed such performance degradation of CLIP-SF in Supplementary Section B.
>
> (b) Regarding performance drops on CIRR and FashionIQ:
>
> We acknowledge that GCL shows small performance drops on scenario-specific datasets like CIRR and FashionIQ. These datasets demand fine-grained and domain-specific understanding (e.g., fashion knowledge), which may not be fully captured by models fine-tuned on general-purpose image-text datasets such as LCS-558K. We want to clarify that our goal with GCL is not to specialize for every dataset but to provide a broadly applicable loss function that supports general multimodal retrieval across diverse settings. For example, as discussed in lines 203–206 of our manuscript, approaches that train with newly generated datasets targeting specific tasks (e.g., datasets for IT2I or T2IT) may work effectively for the targeted tasks but often fail to generalize to other tasks.
>
> **[W2, L] Discussing Limitations**
>
> We agree with the reviewer that including a clear discussion of limitations is important.
> We will clarify that GCL may underperform on scenario-specific retrieval tasks (e.g., CIRR and FashionIQ), particularly in zero-shot settings, as the reviewer rightly noted. While this is one of the limitations of our work, we emphasize that it stems from the training data used, not from the GCL loss itself. GCL is designed to align multiple modalities within a unified representation space, which is beneficial for general-purpose retrieval tasks. However, for optimal performance in domain-specific applications, GCL can serve as an effective initial training stage, followed by additional fine-tuning using task-specific or domain-relevant data. This hybrid strategy preserves the general alignment achieved by GCL while enabling the learning of fine-grained, task-specific features. We will include this limitation and its potential mitigation strategy in the revised manuscript to provide a more balanced perspective on the capabilities of GCL.

---

> > ### Comment · Reviewer_PYSB · 2025-08-09
> >
> > The reviewer appreciates the authors' rebuttal.
> >
> > The reviewer meant to say [12] in the original review but the reviewer thinks it is reasonable to expect the the settings where one of the first papers identified modality gap of CLIP embedding would be carefully revisited in a following-up work that claims mitigating the gap. Specifically, Section 5 in [12], including more serious study on relationship between modality gap measure and downstream tasks, as well as the performance in fairness setup.
> >
> > Given this as well as the incoherent performance on some of the tasks without further justification in the settings like fine-tuning which the author mentioned, the reviewer cannot recommend accept at the moment. But the reviewer believe the manuscript is potentially important for the community once these issues are fully resolved.

---

> ### Author Response · Authors · 2025-08-09
>
> We sincerely thank the reviewer for the thoughtful feedback, the correction regarding the reference to [12], and the valuable suggestions for further strengthening our work. We recognize the importance of revisiting [12], especially Section 5, to better examine the modality gap’s relation to downstream and fairness-related performances, though this is beyond our current scope. We would like to emphasize that our work was built upon prior findings regarding the relationship between the modality gap and downstream task performance, and we will make this connection more explicit. While we are unable to conduct additional experiments at this stage due to the reviewer’s last-moment response, we will incorporate the reviewer’s suggestions into the revised manuscript.
>
> Specifically, we will 1) clearly discuss how our current analyses relate to the findings in [12], 2) elaborate on the connection between modality gap reduction and downstream task performance, and 3) explain more on the performance drop of CLIP-SF under global setting compared to the local setting. We appreciate the reviewer’s recognition of the potential importance of our work to the community and will ensure that the revised version faithfully reflects these constructive suggestions.

---

### Official Review · Reviewer_g4sT · 2025-07-07

**Clarity:** 3
**Significance:** 3
**Originality:** 3
**Rating:** 5
**Confidence:** 3

**Summary:**

The paper considers producing representations all combinations of image and text to improve retrieval from multimodal signals without splitting the modes. The idea is to use a standard contrastive objective for the individual mode embeddings and a joint embedding of the modalities from a paired image and caption dataset; GCL (generalized contrastive learning) also minimizes the alignment between any pairs of mismatched individual embeddings or joint embeddings or across both.

**Questions:**

In table 5, the biggest loss in average performance comes from removing single-mode pairs. This seems to stem mainly from single modal loss in performance. This presents an interesting trade-off between what is stored for retrieval and performance. To elaborate, compare the two cases

1. For each doc, we store a different embedding based on what kind of query comes (retrieve  given one or two modes). Then we can get the best in each column in table 5. This still can be achieved without requiring triplet datasets by dropping appropriate terms in GCL.
2. For each doc, we store the embedding given by the full GCL. You only get the performance of the last column which is not always the best.

I think this is a fundamental trade-off because you cannot align the a bi-modal signal to each separately when there is information in each mode beyond what is explained by the other. Can the authors comment on when and where I would prefer 2 over 1?

A related question is how GCL would/should be adapted in the context of multi-linear contrastive objectives (e.g. [this paper](https://arxiv.org/abs/2411.01053)) that can handle missingness?

**Ethical Concerns:**

["NO or VERY MINOR ethics concerns only"]

**Final Justification:**

Strong paper. No serious concerns original. Discussion clarified a few things.

**Limitations:**

See Weaknesses.

I want to see bootstrap confidence intervals for at least a few results. I do believe the improvements will still be significant but it will help with the ablation study.

**Quality:**

3

**Strengths And Weaknesses:**

- The paper proposes a simple way to build representations for individual modes and also jointly such that there is alignment between any combination of modalities (figure 2 provides empirical intuition)
- The single method provides strong empirical improvement in multimodal retrieval across different datasets and combinations of modes compared to existing methods. The important fact is that the retrieval did not have to get multimodal pairs (image + text 1 associated with image + text 2) to produce these improvements.
- The ablations in section 5 show the value of removing certain terms from the GCL loss which helps uncover a little about where the improvements come from.


Weaknesses
- The space of terms combinatorially explodes as the number of modes increase. There is a lot of image-text multimodal data, so GCL as it stands is still valuable, but I do wonder how training can be easy with even the two modes considered.
- Contrastive learning has an interpretation of learning a classifier that looks at the density-ratio between the joint of two modes and the marginals. That interpretation seems lost with GCL. Can the authors explain?
- The experiments use existing pretrained multimodal retrieval models. Would the method work similarly well if starting from scratch?

---

> ### Author Rebuttal · Authors · 2025-07-30
>
> We sincerely appreciate the reviewer’s thoughtful and encouraging comments. The reviewer found our proposed method to be simple yet effective and appreciated the empirical intuition presented in Figure 2. We also thank the reviewer for highlighting the strong performance of our method across diverse datasets without relying on fully multimodal pairs, as well as the usefulness of the ablation studies in uncovering the contributions of individual loss components. We would like to note that Reviewer h6VU similarly praised the novelty of our loss function and the thoroughness of our experimental analysis. Regarding the concerns raised, we have carefully considered each point and addressed them below. All relevant revisions will be incorporated into the final version.
>
> **[W1] Scalability of GCL with Multiple Modalities and Its Practical Advantages**
>
> We sincerely appreciate the reviewer’s insightful observation regarding the potential combinatorial complexity that may arise in multimodal learning scenarios. This is indeed a valid concern, particularly in settings involving a large number of modalities. However, current approaches in the field require the creation of dedicated datasets for every pairwise or higher-order modality combination, which entails a significant annotation burden. In contrast, while extending GCL to incorporate additional modalities may increase computational cost for the extended similarity map, we believe that it remains comparatively more efficient and practical than constructing new annotated datasets for each modality combination.
>
> A key advantage of GCL is its ability to leverage readily available image-text paired datasets, eliminating the need to manually construct complex triplet-based samples or design specialized datasets for certain retrieval scenarios. Within each mini-batch, GCL automatically generates negative examples by considering all pairwise combinations of image, text, and fused image-text representations. This reduces reliance on labor-intensive data curation and simplifies the training process while preserving the ability to learn across diverse modalities. Moreover, GCL can be readily applied to existing pre-trained cross-modal models without necessitating significant changes to model architectures or training pipelines. This simplicity and flexibility make GCL particularly well-suited for real-world applications, where scalability and ease of integration are critical.
>
> **[W2] Density-Ratio Interpretation in GCL**
>
> We thank the reviewer for this deep theoretical question. We believe that GCL does not discard the density-ratio perspective; rather, it extends it to suit multimodal retrieval with three modalities: image (I), text (T), and fused image-text (IT).
> Traditional contrastive learning estimates the density ratio
> $\frac{p(x, y)}{p(x)p(y)}$
> by contrasting positive pairs against negatives sampled from the marginal distributions. This is implicitly achieved by training a classifier that distinguishes positive pairs from negatives, where the output logits can be interpreted as approximating this density ratio.
>
> GCL generalizes this idea by simultaneously modeling the distribution of the density ratio across six combinations. As shown in Equation 2 of our main paper, for each of these positive pairs $(e^j_a, e^{j}_{b})$, the loss function encourages their similarity to be high relative to all other possible combinations of query $e^j_a$ with any other sample $e^k_m$ from any modality $m \in$ {I,T,IT} within the mini-batch. This inherently means that GCL implicitly estimates a corresponding density ratio for each positive modality pair. From this perspective, we believe that the density-ratio intuition remains preserved in GCL while being extended to a greater number of modality combinations.
>
> **[W3] Effectiveness of GCL from Scratch**
>
> Our experiments utilize pre-trained models to highlight the effectiveness of GCL as a fine-tuning method that enhances existing systems without requiring new data generation. This design choice was made to emphasize the practicality of GCL and its ease of integration. In theory, GCL is also applicable when training from scratch, as it continues to promote consistent alignment across image, text, and fused representations. However, in practice, training from scratch would be considerably more demanding. Pre-trained models such as CLIP already provide strong vision-language representations learned from large-scale datasets like LAION-2B. GCL builds on this solid foundation to further refine alignment across modalities by reducing the modality gap present in such representation spaces. When training from scratch, the model must simultaneously learn both the basic vision-language associations and the more nuanced fused-modal relationships. This would require significantly larger datasets and greater computational resources. Therefore, while training retrieval models with GCL from scratch may be feasible, it introduces practical challenges that are not encountered in our fine-tuning-based approach.
>
> **[Q1] Trade-off Between Modality-Specific vs Unified Embeddings**
>
> As the reviewer correctly pointed out, there exists a trade-off between storing embeddings for each modality separately and using a single unified representation space.
>
> **Strategy 1 (modality-specific embeddings)** is useful when the query modality is known in advance (e.g., a system dedicated solely to image-to-text or text-to-image retrieval). In such cases, it is possible to drop certain GCL terms during training and optimize for specific tasks. This setup aligns with our "local setting" experiments, which show that this approach often yields the best performance for a given task.
>
> **Strategy 2 (embeddings in unified representation space)** is designed for "global setting" use cases where the system must support retrieval with arbitrary types of queries and candidates (image, text, or fused). This is common in real-world systems such as search engines or knowledge base retrieval tools, where the query modality is unknown and can vary at runtime. In this case, having a single, unified embedding for each document greatly simplifies storage and retrieval, since only one index and one retrieval pass are needed regardless of the query type.
>
> While Strategy 2 may show slight performance drops in highly specialized retrieval tasks, it brings major benefits in flexibility, efficiency, and generalization. Therefore, the choice depends on the application: Strategy 1 is best for fixed and known tasks, while Strategy 2 (enabled by GCL) is better for real-world systems requiring universal retrieval across mixed modalities. We will include this valuable insight regarding the use cases of multimodal retrieval in our camera-ready version.
>
> **[Q2] Relation to Symile and Handling Missing Modalities**
>
> When applying GCL to Symile [1], we can generate embeddings for modality combinations that the study does not yet address. Although Symile considers three modalities, image, text, and audio, it omits the pairwise combinations of $it$, $ia$, and $ta$. GCL enables the construction of these embeddings and supports additional retrieval tasks, such as $ita$ $\rightarrow$ $ia$ or $ita$ $\rightarrow$ $i$, which are not included in the evaluation of this paper. We will cite Symile in the related work section of the revised manuscript and highlight how adopting these ideas can broaden the applicability of GCL.
>
> [1] Contrasting with Symile: Simple Model-Agnostic Representation Learning for Unlimited Modalities (NeurIPS 2024)
>
> **[L1] Performance Variance Across Multiple Runs**
>
> We agree that variance estimates strengthen the experimental results. We reran two additional experiments (three in total) for GCL and its ablation study with different random seeds. As shown, we observe a similar tendency to that in the original manuscript, indicating that each tuple of loss term contributes meaningfully to the modality pairs it is designed to supervise. In the supplementary of the revised manuscript, we will report both the mean and standard deviation of performance across these runs.
>
> | Method                                                    | Score         |
> |-----------------------------------------------------------|---------------|
> | GCL w/o ($\mathcal{L}{t2i}, \mathcal{L}{i2t}$)       | 28.09 ± 0.24  |
> | GCL w/o ($\mathcal{L}{t2it}$, $\mathcal{L}{i2it}$)      | 32.85 ± 0.09  |
> | GCL w/o ($\mathcal{L}{it2i}$, $\mathcal{L}{it2t}$)      | 33.61 ± 0.17  |
> | GCL                                                       | 34.03 ± 0.13  |

---

> > ### Comment · Reviewer_g4sT · 2025-08-06
> > **Thanks for the response**
> >
> > The response clarified the few questions I had. I will maintain my current positive score.

---

### Note · Authors · 2025-08-12

We sincerely thank all reviewers and Area Chair for their insightful feedback and careful evaluation. We have thoroughly addressed the raised concerns and provided clarifications as follows:

- Reviewer g4sT recognized the strengths of our work, and we clarified key aspects of GCL’s scalability, theoretical foundation, and practical applicability. We appreciate the continued support, as reflected in maintaining an accept score (5).

- Reviewer PYSB raised important questions about the relation between the modality gap reduction and downstream task improvements and results regarding CLIP-SF. We responded with detailed analyses and acknowledged limitations on specialized datasets, noting that addressing these limitations remains an important direction for future work. We will sincerely clarify these points in the final version.

- Reviewer kEbb acknowledged the important motivation underlying our work. We thoroughly addressed concerns regarding novelty, suitability to NeurIPS, reproducibility, formatting, and computational cost. However, due to the lack of further comments, we were unable to engage in a constructive discussion regarding these issues.

- Reviewer h6VU’s concerns regarding evaluation (e.g., clarification on certain retrieval tasks and test set coverages) and optimization difficulty were fully addressed. We are grateful for the reconsideration, which led to a score increase from borderline accept (4) to accept (5).

Overall, these interactions have strengthened our manuscript and helped provide a comprehensive response to reviewer concerns. We thank the reviewers and the Area Chair for their time and valuable input, which have greatly improved the clarity and rigor of our work.

---

### Decision · Program_Chairs · 2025-09-17

**Decision:**

Accept (poster)

**Comment:**

The authors propose Generalized Contrastive Learning (GCL), which extends the standard contrastive learning formulation to the multi-modal setting by using the contrastive objective with negative samples being all possible combinations of embeddings (e.g., image embeddings, text embeddings, fused embeddings) and positive samples being samples with different modalities from the same pair (i.e., contrastive learning is performed across all possible combinations). The resulting embedding is designed to improve universal multimodal retrieval without requiring constructing complete datasets (i.e., containing examples aligned across modalities) and can use existing datasets. Using GCL to augment VISTA, CLIP, and TinyCLIP, the authors demonstrate consistent performance improvements on subsets of widely used multimodal retrieval benchmarks (M-BEIR, MMEB, CoVR) over baseline models and those finetuned with generated data triplets.

Strengths identified by the reviewers includes:
- GCL is a simple, yet effective way to construct embeddings for individual modalities in a joint space such that there is alignment between any combination of modalities. GCL is conceptually intuitive and doesn't require generating specialized datasets. Both the backbone model and datasets are essentially off-the-shelf.
- There has been a recent increased interest in multimodal embeddings, particularly in post-training LLM scenarios.
- GCL demonstrates strong empirical performance on relevant datasets against relevant baselines.
- The secondary experiments (e.g., ablations) help understand the relative contribution of different innovations and the overall dynamics of GCL.
- The paper is well-written and has a sufficient number of experiments to validate the performance for text-image retrieval settings (and pooled image embeddings for video settings).

Weaknesses identified by the reviewers includes:
- GCL is presented as a general multimodal embedding, but computing all combinations of modalities may be challenging at high-cardinality. However, in most practical settings, there is likely to be <5 modalities, so this isn't a practical concern (although maybe a bit of an over-claim). As stated in rebuttal, existing methods would likely require more effort in constructing the training data.
- In this vein, experiments with additional modalities would (obviously) strengthen the demonstrated utility of GCL.
- While novel, this work can be interpreted as a incremental innovation over existing multi-modal embedding papers and may be of primary interest to a more narrow community (e.g., multimodal since single-mode embeddings may perform better for single-mode tasks). Specifically, the authors are not defining a new problem nor providing new datasets, etc. -- so it is incremental work methodologically. While the formulation is general, the instantiation in this paper is more narrow.
- There were several more theoretical/analytical questions, possibly the most interesting being the relationship between the modality gap and downstream performance (reviewer PYSB). However, these were well addressed in rebuttal -- even if it would be good to see a version of the paper with these rebuttal incorporated into the main text.
- GCL doesn't outperform competing models is all experimental settings (e.g., FashionIQ), but this isn't surprising and just needs to be discussed in the paper.

Overall, as the authors state, GCL is a simple but effective method for constructing multi-modal embeddings. The experiments are exhaustive and sufficiently convincing for the text-image/video retrieval setting. The paper would be stronger with more modalities (likely not possible in this revision) and some of the theoretical and clarification discussion raised by the reviewers (likely possible as these were sufficiently addressed in rebuttal) -- especially since the method is so 'simple', more rigorous analysis is possible. The reviewers that engaged with the authors during rebuttal raised their scores (including h6VU which intended to do so) and the reviewers that did not had their concerns adequately addressed in my opinion. Thus, I view the paper as borderline, but lean toward accepting contingent on competing papers.